# Five-Surface Phosphor-in-Glass for Enhanced Illumination and Superior Color Uniformity in Large-View Scale LEDs

**DOI:** 10.3390/mi15080946

**Published:** 2024-07-24

**Authors:** Hong-Wei Huang, Chien-Wei Huang, Yi-Chian Chen, Hsing-Kun Shih, Wei-Chih Cheng, Chun-Nien Liu, Chia-Chin Chiang

**Affiliations:** 1Department of Mechanical Engineering, National Kaohsiung University of Science and Technology, Kaohsiung 807, Taiwan; weialone@gmail.com (H.-W.H.); ccchiang@nkust.edu.tw (C.-C.C.); 2Department of Electrical Engineering, National Chung Hsing University, Taichung 402, Taiwan; cwhuang6374@dragon.nchu.edu.tw (C.-W.H.); d108064302@mail.nchu.edu.tw (H.-K.S.); weichih0428@gmail.com (W.-C.C.); 3Department of Occupational Safety and Hygiene, Fooyin University, Kaohsiung 831, Taiwan; 4Graduate Institute of Optoelectronic Engineering, National Chun Hsing University, Taichung 402, Taiwan

**Keywords:** five-surface phosphor layer, phosphor-in-glass, laser engraving process, chip scale package, white light-emitting diode

## Abstract

A novel five-surface phosphor-in-glass (FS-PiG) structure for high illumination and excellent color uniformity in large-view scale LEDs for sensor light source application is demonstrated. YAG phosphor (Y_3_Al_5_O_12_:Ce^3+^) was uniformly mixed with ceramic and sintered at 680 °C to form a phosphor wafer. Sophisticated laser engraving was employed on the phosphor wafer to form saddle-shaped large-view scale FS-PiG LEDs. The performance of the FS-PiG LEDs exhibited an illumination of 401 lm, average color temperature (CCT) of 5488 K ± 110 K, and color coordinates (CIE) of (0.3179 ± 0.003, 0.3352 ± 0.003). In contrast to convention single-surface phosphor-in-glass (SS-PiG) LEDs, the performance exhibited an illumination of 380 lm, average CCT of 5830 K ± 758 K, and CIE of (0.3083 ± 0.07, 0.3172 ± 0.07). These indicated that the performance of the FS-PiG LEDs was higher than the SS-PiG LEDs for illumination, CCT, and CIE by 1.7, 7, and 23 times, respectively. Furthermore, the FS-PiG LEDs demonstrate a lower lumen loss of 2% and a reduced chromaticity shift of 5.4 × 10^−3^ under accelerated aging at 350 °C for 1008 h, owing to the high ceramic melting temperature of up to 510 °C. In this study, the proposed FS-PiG large-view scale LEDs with excellent optical performance and high reliability may be promising candidates to replace the conventional phosphor-in-organic silicone material used in high-power LEDs for the next generation of sensor light sources, display, and headlight applications.

## 1. Introduction

White light-emitting diodes (LEDs) are essential because they can replace traditional light sources, such as incandescent and fluorescent lamps, due to their high luminous efficiency, long lifetime, and environmental friendliness [1,2]. One major challenge in white LED technology is achieving high color uniformity and better reliability, which is critical for high-power lighting applications. There are some essential approaches for attaining high color uniformity in white LEDs. One uses the scattering effect to deflect the light direction to balance blue and yellow light distributions by introducing micro- and nano-particles/structures and optimizing the phosphor particle size [3,4,5,6]. The other uses the Beer–Lambert law to adjust the amount of blue light at specific angles by optimizing the thickness distribution, location, and material of phosphor-converted layers [7,8,9,10]. However, there are still limited technologies for achieving high color uniformity (CCT and CIE) and good reliability in large-scale white LEDs. Therefore, developing a suitable large-scale white LED package is essential to achieve high color uniformity with excellent reliability. 

Furthermore, the LED modules’ low light extraction efficiency (LEE) has been an important issue. This is due to the total internal reflection (TIR) effect [11] on the top encapsulation surface of traditional modules, which reduces light emission and leads to the conversion of captured light rays into heat. Researchers have recently developed various methods to reduce TIR and enhance LEE. These include nano-/micro-scale surface structures [12], microlens arrays [13], and precision mechanical engraving structures [14]. However, nano-/micro-scale approaches may be costly. The microlens arrays may also suffer from optical dispersion and material issues, affecting uniformity and causing damage, while precision mechanical engraving is limited to millimeter-scale sizes due to equipment constraints. The laser engraving approach is a promising technology for creating microstructures on ceramic phosphor plates without contact or chemical agents to achieve high accuracy, small dimensions, and the ability to shape and pattern as desired. In laser engraving, the pulsed picosecond lasers enable the adjustment of processing depth through power modulation and the processing area through beam diameter adjustment, to create intricate patterns.

The primary preparation for the chip-scaled package–white light-emitting diode (CSP-LED) is as follows. The CSP-LED processes involve two parts: pressing chip arrays with yellow phosphor-in-organic silicone material (PiOS) and cutting them into a single LED. The substrate prepares LED chips and applies colloids before placing the yellow PiOS material sheet on the chip. Simultaneous hot pressing combines the phosphors on the chips. The resulting wafer is cut into single LEDs. There are many defects in the pressing process with the PiOS, such as the poor uniformity of luminous intensity, uneven heat transfer, and poor color temperature consistency. The poor thermal stability of the PiOS may be due to the lower silicon transition temperature of 150 °C. These defects can lead to a lower product rate, short life, etiolation, and apertures with noticeable brightness differences in the lighting range.

Regarding packaging structure, it is directly attached to the chip surface and can achieve high color uniformity through its sufficient scattering ability and uniform thickness distribution. The manufacturing methods used were pulse jetting [15], dispensing [16], compression molding [8], or 3D printing [17], to control the top thickness of the encapsulant, forming chip-scale packaging (CSP) structures. These CSP structures have the potential to achieve both a small size and a high luminous density.

In this study, we propose a novel FS-PiG for high illumination and excellent color uniformity in large-view scale LEDs, specifically targeting applications with an extended field of view (FoV) for sensor light sources. The FS-PiG structure is the CSP-LEDs. The structure utilizes a pentahedron to cover the LED light source, enhancing the contact between the light source and the effective phosphor area compared to a flat surface. Moreover, it allows for controlling the light emission angle to concentrate the light more effectively. YAG phosphor (Y_3_Al_5_O_12_:Ce^3+^) was uniformly mixed with phosphor in glass and sintered at 680 °C to form a PiG wafer. The laser engraving was employed to manufacture a saddle-shaped FS-PiG structure on the surface of the LED wafer. Then, we packaged the saddle-shaped FS-PiG with the LED flip chip. The experiment aimed to create a micro-square via structure on the ceramic phosphor plates by laser ablation. We expect the micro-square via structure to enhance the ceramic phosphor plates’ light extraction and scattering properties, thereby improving their luminous performance.

The FS-PiG-based white LEDs demonstrate a lumen loss of only 2% and a reduced chromaticity shift of 5.4 × 10^−3^ under accelerated aging at 350 °C for 1008 h, owing to the high ceramic melting temperature of up to 510 °C. Unlike conventional conformal package structures, the top and side thicknesses of FS-PiG are controllable and show only a 0.01% CCT shift at 6000 K, exhibiting excellent potential to achieve high color uniformity. The proposed FS-PiG larger-view scale LEDs with outstanding optical performance and high reliability may be promising candidates to replace the conventional phosphor-in-organic silicone material used in high-power LEDs for the next generation of sensor light sources, display, and headlight applications.

## 2. Fabrication

The fabrication process of the FS-PiG structure for the CSP-LED involves several critical steps. First, achieving the uniform mixing of the phosphors with the ceramic powder by optimizing the mixing time, temperature, and pressure is crucial. Second, the sintering temperature and time are essential for forming the PiG wafer, as they affect the densification and homogeneity of the phosphor–ceramic composite. Third, the laser processing parameters, such as laser power, scanning speed, and pulse frequency, need to be carefully controlled to ensure the formation of a uniform and precise saddle-shaped FS-PiG structure. Finally, we must optimize the packaging process for an excellent thermal and mechanical connection between the FS-PiG and the LED flip chip (BGB 1 R45 A, 450 nm of InGaN Blue LED Chip, Huaian Aucksun Optoelectronics Technology Co., Ltd., Huaian, China).

The compositions of the glass matrix were 10 mol% B_2_O_3_, 10 mol% Sb_2_O_3_, 70 mol% SiO_2_, and 10 mol%Ta_2_O_5_. The mixed raw materials were melted at 557 °C. After cooling, we ground the glass matrix of the B_2_O_3_-Sb_2_O_3_-SiO_2_-Ta_2_O_5_ (D263, Schott AG, Jena, Germany) into ceramic powders with a size of about 10 μm. Next, we uniformly spread the yellow phosphors (Y_3_Al_5_O_12_:Ce^3+^, NYAG4454, Intematrix Corp., Fremont, CA, USA) with a size of about 15 μm into the glass matrix powder with a weight ratio of 84 wt%. The absorption and luminescence excitation spectra were at 450 nm and 540 nm. The average powder size distribution was 0.34 um, as shown in Figure 1. The material was mixed with a tubular vibration mixer for 4 h at 450 rpm. We then pressed the mixture powder to form a PiG precursor. Then, a 4 inch PiG powder block was produced and sintered at 680 °C for 30 min and annealed at 350 °C for 120 min below the glass transition temperature in a ceramic die-casting furnace. In the cutting and grinding process depicted in Figure 2, we controlled the thickness and diameter of the PiG wafer, maintaining them at 0.3 mm and 90 mm, respectively.

Laser engraving was employed to create precise micro-patterns on phosphor films within PiG wafers; the shape was a square and concave structure. Then, UV picosecond laser screw-feed processing was used to modify the PiG wafer surface. Then, we manufactured a saddle-shaped FS-PiG structure through high-temperature annealing and acid-etching processes. We adjusted parameters such as laser power, frequency, and focus throughout fabrication to achieve the desired FS-PiG structure on the PiG wafer. These experimental parameters significantly affected the material removal rate, surface quality, and morphology of the FS-PiG structures. Laser power was crucial in controlling the material removal rate and depth. While higher power facilitated more significant material removal, it also posed the risk of thermal damage and micro-cracks. Therefore, optimizing the laser power was necessary to ensure the quality of the FS-PiG structure. Laser frequency, however, influenced the size and shape of the FS-PiG structures. Determining the optimal frequency was vital to achieving the desired microporous structure. During the laser engraving process, we needed to consider bottom flatness, opening chipping, and processing slope, as shown in Figure 3. The laser engraving processing parameters are 44 W laser energy, 1.3 m/s scanning speed, and 33 s processing time—more detailed parameters are listed in Table 1. Figure 4 shows each fabricated saddle-shaped FS-PiG structure with an inside/outside width of 1.25/1.5 mm and a depth of 0.2 mm, respectively, which would fit the LED chip size of 45 mil. We used an optical coherence tomography device (IVS-2000-HR, Santec, Japan) to measure the information about the surface roughness, uniformity, and topographical characteristics of the PiG layer, as shown in Figure 4c. These images can reveal grain size, particle distribution, and surface texture, visually representing the phosphor layer’s morphology. Because of the rough surface of the saddle-shaped FS-PiG structure, it was difficult to bond with the LED chip. The secondary annealing improved the surface roughness of this microstructure. In this study, the saddle-shaped FS-PiG structure was annealed in air for 12 h at a temperature of 300 °C to smooth the inside of the structure surface.

Epoxy curing was used to create the white CSP-LEDs for the package between the prepared saddle-shaped FS-PiGs and the blue LEDs. The CSP-LED is necessary to ensure alignment, adhesive dispensing parameters, and uniform coating during packaging. The FS-PiG chip was bonded by a controlled pressure using an X-Y axis platform, adjusting pressure and temperature settings. The following steps outline the detailed implementation process:Step 1:Positioning each FS-PiG structure and the blue LED array with the substrate ensures the surface is clean and dust-free to prepare the CSP.Step 2:Utilize a dispenser to apply translucent epoxy glue to each blue LED to ensure the precise control of the glue’s amount and placement.Step 3:The FS-PiG structure can be picked up using the vacuum nozzle and then transferred to the appropriate position before being placed onto the blue LED. Then, the pressing machine forms a sealed cavity between itself and the platform of the operating table, and it pumps the cavity into a vacuum, with a vacuum degree ranging from −95 to 100 kPa.Step 4:We maintained a fixed pressure of 20–30 kgf/cm^2^ and pressed for 5–10 min with the pressing platform.Step 5:We put the blue LED-covered FS-PiG chip into the oven for sintering and curing to enable the adhesive to solidify. We maintain the sintering temperature at 150 °C, usually for about 2 h. To achieve the complete curing of the epoxy resin, we maintained the curing temperature at 120 °C for 4 h, as shown in Figure 5.

The abovementioned methods could be used to package the chip, emitting light through five chip surfaces. The top and side surface thicknesses are 100 μm and 125 μm, respectively. The saddle-shaped FS-PiG thickness of the four sides of one chip is the same. Figure 6 shows the CSP-LED with FS-PiG on the substrate, which is just for testing light performance more conveniently. 

## 3. Results and Discussion

### 3.1. Analysis of the Laser Processing

In this study, we prepared ceramic phosphors based on Y3Al5O12:Ce3+ by the heated press-sintering method [18]. We investigated their photoluminescence properties and performance as converters for warm-white LEDs with a high color-rendering index. To evaluate the optical and thermal properties of the FS-PiG-based white LEDs, we measured their luminous flux, color temperature, and chromaticity coordinates at different operating currents. In addition, we performed accelerated aging tests on the LEDs to investigate their reliability under high-temperature conditions. The experimental results have shown that these parameters produce a high-quality structure with precise and uniform features, which can provide excellent thermal conductivity and package properties for CSP packaging. 

These techniques provide valuable information about the surface roughness, uniformity, and topographical characteristics of the phosphor layer, as shown in Figure 7 and Figure 8. The images can reveal grain size, particle distribution, and surface texture, visually representing the phosphor layer’s morphology. We performed a laser engraving process on the surface of a five-layer phosphor layer, which carves a concave shape into the fluorescent substrate. We achieved a depth of 132 um with the laser, creating a precise laser micro-via structure with a diameter of 1.5 mm, as shown in Figure 8c. The laser engraving resulted in a well-defined and controlled surface morphology on the phosphor layer. The deep laser engraving reached a depth of 132 um, allowing for the creation of intricate patterns and structures within the phosphor layer. This deep laser machining capability opens up possibilities for various applications that require precise and profound modifications within the phosphor layer.

Additionally, the laser engraving created during the process had a diameter of 1.5 mm. These laser micro-via structures serve as channels or pathways for light propagation within the phosphor layer. The precise and controlled formation of these micro-via structure ensures efficient light transmission and distribution, contributing to the overall performance of the phosphor layer.

### 3.2. Optical Characterization

The luminous flux of the FS-PiG-based white CSP-LEDs was measured by using an integrating sphere system (LMS-8000, LISUN Instruments Limited, Shanghai, China) at different driving currents ranging from 350 mA to 1000 mA, as shown in Figure 9. The luminous flux of the LEDs increases with increasing driving current. At a driving current of 1000 mA, the device achieves a maximum luminous flux of 401 lm. The maximum luminous flux of FS-PiG was 1.05 times larger than SS-PiG when the maximum luminous flux of SS-PiG was 380 lm.

The color temperature and chromaticity coordinates of the FS-PiG-based white CSP-LEDs were measured using a spectroradiometer (CAS 140CT, Instrument Systems, Munich, Germany) equipped with an integrating sphere (IS-1.5-SA, Instrument Systems, Munich, Germany) at different driving currents. The color temperature of the LEDs increases from 5638 K to 5935 K as the driving current increases from 350 mA to 1000 mA. The chromaticity coordinates of the LEDs at different driving currents are shown in Figure 9. Observers can note that the chromaticity coordinates remain almost constant at (0.35, 0.39) within the entire current range, indicating the excellent color stability of the FS-PiG-based white CSP-LEDs. We calculated the conversion efficiency of the phosphor in the CSP-LED, as it is necessary to determine the input power of the LED. The constant forward voltage was 3.2V for the CSP-LED, and the conversion efficiency can be measured using luminous efficiency. This estimation assumes that the light source’s internal quantum efficiency (IQE) is close to 100% and that the phosphor maintains a constant spectral power distribution (SPD) at all current levels. In this study, we approximate the conversion efficiency as the ratio of the luminous flux emitted by the LED with the micro-square via structure to the luminous flux emitted by the LED without the micro-square via structure, both operating at the same current level. Assuming that the LED without the micro-square via structure has a luminous efficacy of 200 lm/W, the conversion efficiency can be calculated at different current levels as follows:At 350 mA: Efficiency = (199.40 lm/0.1778 W)/(350 mA × 3.2 V/0.2) = 87.4%
At 500 mA: Efficiency = (254.74 lm/0.1592 W)/(500 mA × 3.2 V/0.2) = 84.5%
At 700 mA: Efficiency = (318.92 lm/0.1422 W)/(700 mA × 3.2 V/0.2) = 83.6%
At 1000 mA: Efficiency = (401.43 lm/0.1255 W)/(1000 mA × 3.2 V/0.2) = 82.9%

Thus, we calculated the conversion efficiency of these FS-PiG-based white CSP-LEDs to be about 83–87%.

For the electroluminescence spectra experiment, we used a spectroradiometer to measure the radiation intensity of CSP-LEDs with different wavelengths and currents. To investigate the photoluminescence (PL) spectra of the fabricated FS-PiG-based white CSP-LEDs, we used a spectrometer (QE65000, Ocean Optics, Orlando, FL, USA) to measure the EL spectra. We measured the PL spectra at different driving currents, ranging from 50 mA to 1 A. Figure 10 presents the PL spectra of the FS-PiG-based white and conventional LEDs at a driving current of 1 A. The PL spectra of the FS-PiG-based white LEDs are broader than those of the traditional LEDs, indicating that the FS-PiG layer can effectively broaden the PL spectrum. Using a micro-square, the PL spectrum can be measured using a spectrometer to analyze the emitted light from an LED. We can adjust the current to drive the WLED lamp from 350 mA to 1000 mA. By examining the PL spectrum, we could identify the peak emission wavelength (PWL) of the phosphor used in the LED with a micro-square via.

The intensity of the CSP-LED is wavelength-dependent, as evidenced by the varying emission spectra. The measurement repeatability of the intensity was evaluated by performing multiple measurements under the same conditions. The repeatability was calculated using the standard deviation and expressed as the mean value. The performance of CSP-LEDs depends on several factors, such as the thickness and concentration of the phosphor film on the top and side surfaces of the chip and the current density of the CSP-LEDs in Figure 11a,b. The measurement results of the micro-square via-structure phosphor with a blue LED showed a luminance intensity of 275.

In comparison, the blue LED without the structure had a measured luminance intensity of 160. The light intensity of the five-surface structure was higher by 1.7 times. These results indicate that the photoluminescent properties of the phosphors are significantly improved when combined with the micro-square via structure. The micro-square via structure enhances the light extraction efficiency and directs the emitted light in the desired direction. This structure allows more light from the phosphor to escape from the LED chip, increasing the luminous intensity.

The luminous flux measures the perceived power of light emitted by a source. With the increase in thickness and concentration of the phosphor film, the luminous flux of CSP-LEDs decreases because the phosphor layer absorbs more blue light, resulting in the emission of less white light. The ratio of the top surface film thickness to the side surface film thickness affects the light distribution curve. However, decreasing the film thickness too much can also reduce the luminous flux due to the incomplete conversion of blue light into white light. The YAG: Ce^3+^ phosphor with the microstructure exhibited higher photoluminescence intensity under blue light excitation than the phosphor without the microstructure. The main factor for this is that the microstructure enhances the light scattering and absorption of the phosphor, thereby enabling a more efficient transfer of energy from blue light to Ce^3+^ ions. However, the effect makes the performance of the FS-PiG LED on the color coordinate similar to the white light standard at (0.33, 0.33) when the color coordinates in the five-surface structure and the traditional type were (0.3179 ± 0.003, 0.3352 ± 0.003) and (0.3083 ± 0.07, 0.3172 ± 0.07). The offset in the SS-PiG was higher than 2.6 times with the FS-PIC and was similar in the blue light area.

Figure 12 shows the advantages of the emitting light type in the different emitting angles, which compares color coordinate distribution, CCT, and luminous performance. Figure 12a shows the X-axis profile analysis of EIC in the SS-PiG and FS-PiG. The center points of SS-PiG and FS-PiG were similar and close at 0.31 and 0.32, but the standard deviation gap was 23 times greater when the values were 0.07 and 0.003, respectively.

At the same time, the above trends are also reflected in color temperature at various emitting angles. As seen in the color temperature measurement in Figure 12b, SS-PiG LEDs have a color temperature range from 5000~7000 K due to their structure, with an average color temperature of 5800 K. Still, the overall color temperature ranges from neutral-color light (3300~5300 K) to cold-color light (above 5300 K). The neutral-color temperature area accounts for 36%, and the cold-light area accounts for 64%, which is biased toward the cold-light area in Figure 11b. The average FS-PiG is 5500 K, at the junction of neutral and cold colors. The deviation value of FS-PiG in 110 K is more uniform than the SS-PiG in 758 K. Finally, the light-emitting area of FS-PiG was three times larger than the SS-PiG when we used a laser engraving process for the PiG to engrave the five-surface structure and induce a more light-emitting area. 

The advantages of five-surface emitting white-light LEDs in terms of light uniformity, softer illumination, and broader lighting effects in optical measurement and sensor applications are as follows:Improved precision: Uniform light distribution can enhance measurement accuracy in optical measurements. The uniform light generated by FS-PiG LEDs helps reduce measurement deviations caused by uneven light spots or shadows.Reduction of deviations: Softer light can reduce light reflection and refraction, thereby minimizing optical deviations that may occur in sensors, enhancing measurement accuracy and reliability.Increased coverage: Due to the broader illumination range, FS-PiG LEDs can cover larger sensing areas, expanding the application range of sensors to accommodate a wider variety of objects of different sizes and shapes.Reduced calibration requirements: Uniform and soft lighting helps reduce the calibration needs of sensors. Sensor calibration becomes more accessible and accurate with a more stable and consistent light environment for measurements.

Based on these advantages, the benefits of five-surface emitting white-light LEDs in optical and sensor applications can enhance measurement accuracy, reliability, and stability, thereby enhancing sensor systems’ performance and application value. Based on the literature [19], using LEDs instead of incandescent lights as the calibration source to calibrate the photometer for measuring LED illumination can significantly reduce the uncertainty associated with spectral mismatch correction. Switching from incandescent lamps to LED calibration sources reduces the average maximum spectral mismatch error of LED measurements by a factor of three.

To evaluate the color performance of the fabricated FS-PiG-based white LEDs with CSP, we measured the color coordinates using a chromaticity meter (CS-200, Konica Minolta, Tokyo, Japan). We measured the color coordinates at different driving currents, ranging from 50 mA to 1 A. Figure 13 shows the FS-PiG-based white and conventional LEDs’ chromaticity coordinates at different driving currents. The chromaticity coordinates of the FS-PiG-based white LEDs are stable at different driving currents, indicating that the FS-PiG layer has excellent color stability. Moreover, the chromaticity coordinates of the FS-PiG-based white LEDs are closer to the standard illuminant D65 than those of the conventional LEDs, indicating that the FS-PiG-based white LEDs have better color rendering.

The FS-PiG possesses a wider luminous angle, and the design structure described in this paper provides a broader emission angle characteristic. The luminous intensity distribution curves for the FS-PiG and SS-PiG were measured using a goniophotometer (LID-100CS, AMA Optoelectronics Inc., Taoyuan, Taiwan), illustrated in Figure 14, highlighting the functional versatility of these phosphor-in-glass structures. The FS-PiG beam pattern appears elliptical due to its larger beam angle. In contrast, the SS-PiG exhibits a more focused and symmetrical beam pattern owing to its smaller beam angle. Specifically, the FS-PiG’s photometric distribution curve demonstrates that its light beam spreads over a range of −62° to 62°, whereas the SS-PiG’s light beam is primarily concentrated between −49° and 49°. The field-of-view angle of the FS-PiG is increased by 27% compared to the SS-PiG. This difference indicates that the FS-PiG provides a more uniform light distribution over a wider angular range. These characteristics suggest that the FS-PiG is more suitable for applications requiring wide-angle illumination, such as outdoor or large-indoor-area flood-lighting.

### 3.3. Thermal Stability

To evaluate the thermal stability of the fabricated FS-PiG-based white CSP-LEDs, we conducted accelerated aging tests at a temperature of 350 °C for 1008 h. We evaluated the thermal stability by measuring the optical output power and color coordinates before and after the accelerated aging tests. We observed that the measured optical output power and color coordinates of the FS-PiG-based white LEDs and conventional LEDs before and after accelerated aging tests did not change the content. The FS-PiG-based white LEDs exhibited lumen loss of only 2% and a reduced chromaticity shift of 5.4 × 10^−3^—values much lower than those of conventional LEDs. The high ceramic melting temperature of up to 510 °C contributes to the good thermal stability of the FS-PiG. To evaluate the thermal performance of the FS-PiG-based white LEDs, we performed accelerated aging tests on the LEDs by placing them in an oven at 350 °C for 1008 h. This specification is based on JESD22 Method A108-C [20]. We monitored the LEDs’ lumen maintenance and chromaticity shift during aging. After 1008 h of aging at 350 °C, the lumen maintenance of FS-PiG-based white LEDs exceeds 98%, demonstrating remarkable thermal stability. The chromaticity shift during the aging process is minimal, with the LEDs’ coordinates shifting slightly from (0.32, 0.33) to (0.33, 0.34), corresponding to a chromaticity shift of 5.4 × 10^−3^. The high ceramic melting temperature of up to 510 °C contributes to the good thermal stability and low chromaticity shift of FS-PiG-based white LEDs, preventing the phosphor from decomposing and diffusing into the epoxy resin under high-temperature conditions. The aging test process is shown in Figure 15.

Based on the measurement above results, laser engraving can provide the potential for mass production with reliable precision. In contrast to 3D printing methods, which offer centimeter- or millimeter-level accuracy [17], laser engraving provides superior control over the structure of fluorescent sheets due to its micron-level precision advantage. Compared to methods such as dispensing [16] or compression molding [8], our approach offers superior control over the structure and thickness of the phosphor material. The high-precision engraving ensures consistent optical characteristics across individual white LEDs, enabling stable optical performance in large-area illumination when employed in multi-module configurations.

## 4. Discussion and Conclusions

In summary, we have demonstrated a novel FS-PiG structure for chip-scale package white LEDs that exhibits excellent optical performance and high reliability. We performed the laser engraving process on the surface of an FS-PiG layer. The laser engraving resulted in a well-defined and controlled surface morphology on the phosphor layer, creating intricate patterns and structures within the phosphor layer. This deep laser engraving capability opens up possibilities for various applications that require precise and profound modifications within the PiG layer. The saddle-shaped FS-PiG structure can achieve high color uniformity with controllable top and side thicknesses, and the yellow phosphor in the ceramic composite has a high ceramic melting temperature of up to 510 °C, which provides excellent thermal stability for the LEDs.

The performance of the FS-PiG LED exhibited an illumination of 401 lm, average color temperature (CCT) of 5488 K ± 110 K, and color coordinates (CIE) of (0.3179 ± 0.003, 0.3352 ± 0.003). In contrast to the single-surface phosphor-in-glass (SS-PiG) structure, it exhibited an illumination of 380 lm, average CCT of 5830 K ± 758 K, and CIE of (0.3083 ± 0.07, 0.3172 ± 0.07). These values indicate that the performance of the FS-PiG was higher than the illumination, CCT, and CIE values for the SS-PiG by 1.1, 7, and 23 times, respectively. The FS-PiG-based white LEDs exhibit lumen loss of only 2% and a reduced chromaticity shift of 5.4 × 10^−3^ chromaticity shift after 1008 h of aging at 350 °C, indicating excellent thermal stability and reliability. The proposed FS-PiG-based white LEDs with CSP are promising candidates to replace the current phosphor-in-organic silicone in high-power LEDs for the next generation of illumination, display, and headlight applications.

## Figures and Tables

**Figure 1 micromachines-15-00946-f001:**
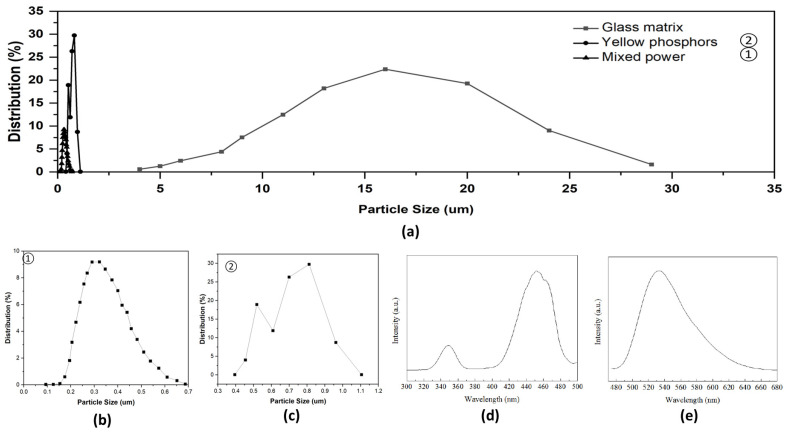
The measurement of single/mixed powder distribution, absorption spectrum, and luminescence excitation spectrum with powder (MALVERN Mastersizer 2000). ■ is the yellow phosphors powder, and the average size is 13.5 um in the figure (**a**); ● is the glass matrix powder, and the average size is 0.7 um in the figure (**b**); ▲ is the mixed powder mixer for 4 h at 450 rpm, and the average size is 0.34 um in the figure (**c**). The absorption spectrum and luminescence excitation spectrum were measured using PL in 450 nm and 540 nm in figure (**d**) and (**e**), respectively.

**Figure 2 micromachines-15-00946-f002:**
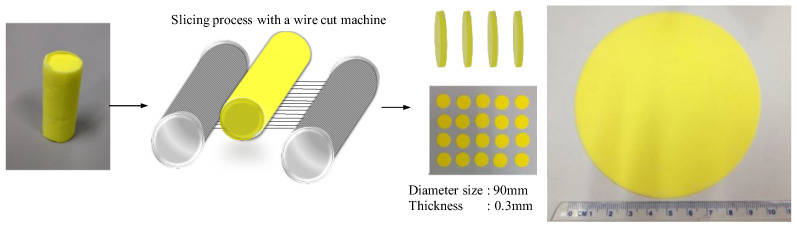
Schematic diagram of PiG wafer cutting and grinding process.

**Figure 3 micromachines-15-00946-f003:**
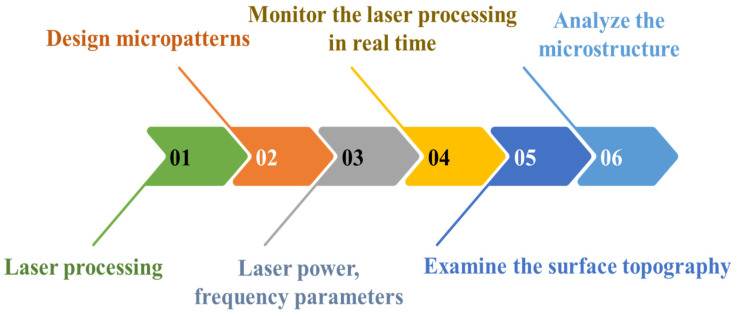
Schematic diagram of laser screw-feed processing.

**Figure 4 micromachines-15-00946-f004:**
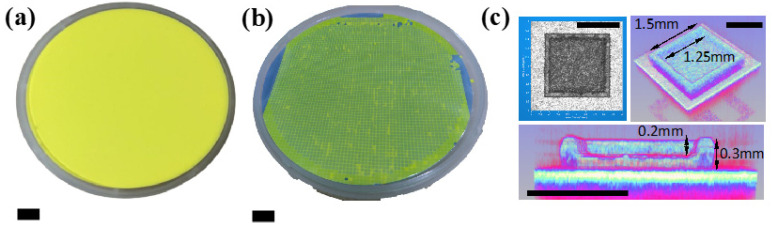
(**a**) PiG wafer (scale bar, 10 mm), (**b**) FS-PiG wafer (scale bar, 10 mm), (**c**) FS-PiG structure (scale bar, 1 mm).

**Figure 5 micromachines-15-00946-f005:**
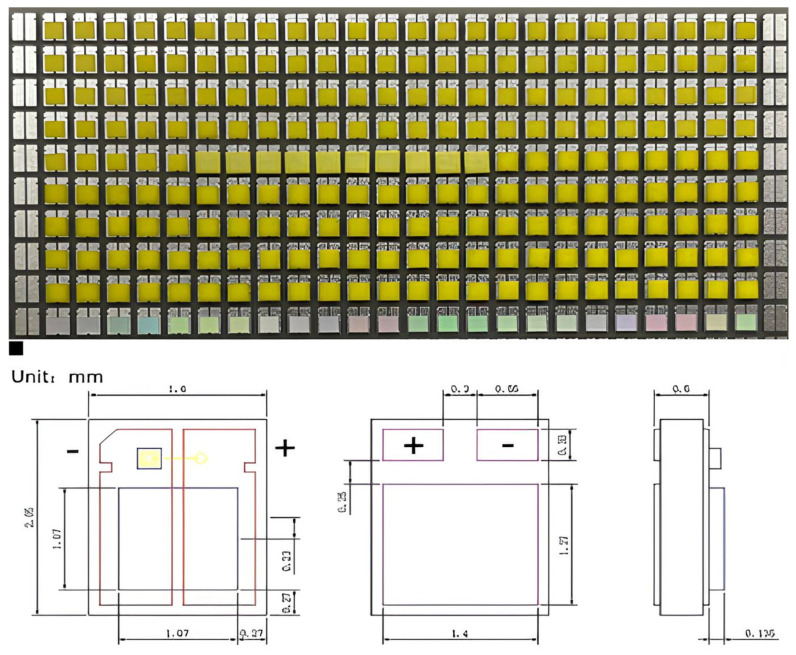
FS-PiG of CSP-LED array board (scale bar, 1 mm) and LED chip size.

**Figure 6 micromachines-15-00946-f006:**
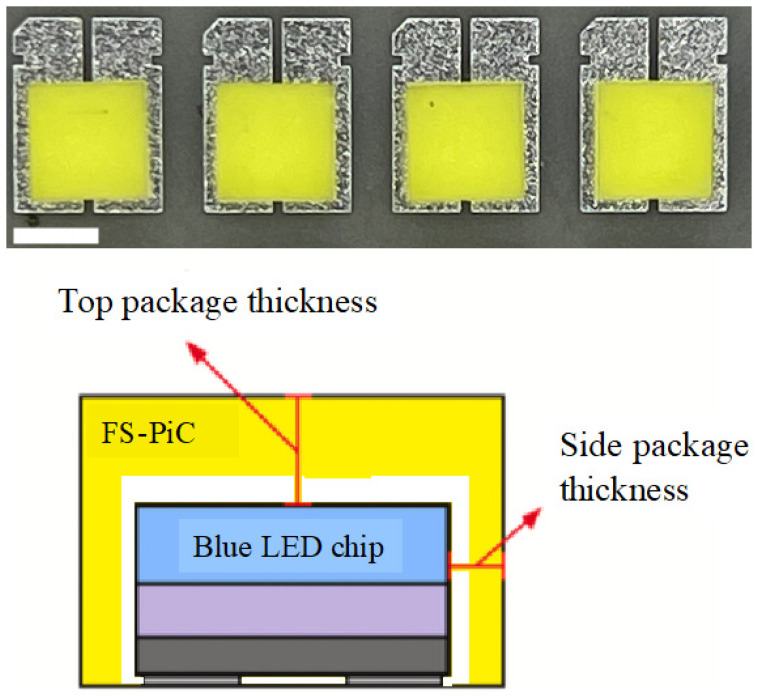
A test sample of CSP-LED. An FS-PiG is coated on a blue LED chip (scale bar, 1 mm), where its light interacts with the phosphor upon diffusion, resulting in absorption/emission, creating a white light source. The top chip diagram shows the result after coating the blue LED chip with phosphor powder.

**Figure 7 micromachines-15-00946-f007:**
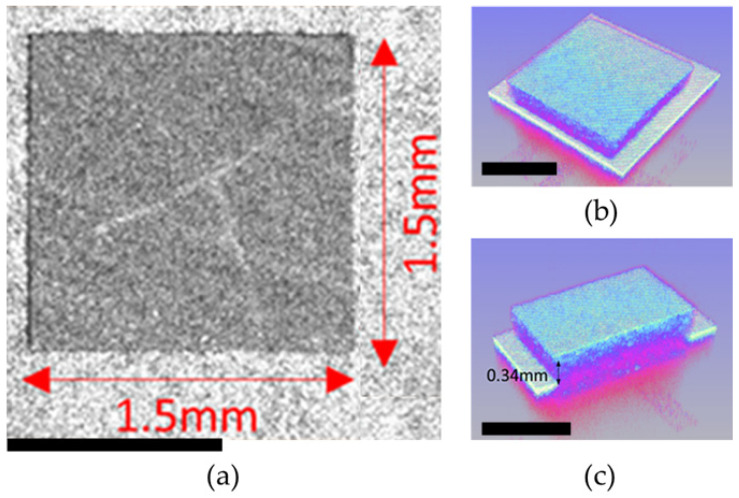
Optical image analysis of PiG (scale bar, 1 mm): (**a**) top view in the 2D image, (**b**) tilt angle in the 3D image, and (**c**) sectional view in the 3D image.

**Figure 8 micromachines-15-00946-f008:**
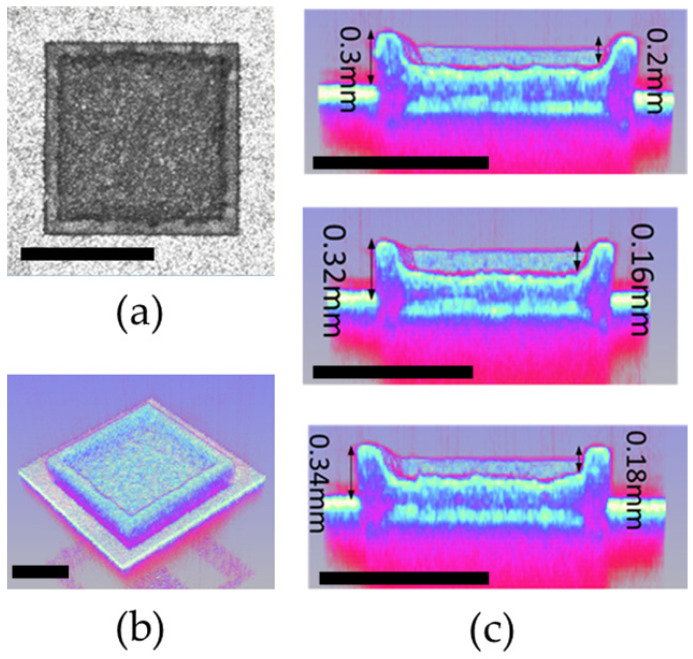
Optical image analysis of FS-PiG (scale bar, 1 mm): (**a**) top view in the 2D image, (**b**) tilt angle in the 3D image, and (**c**) sectional view in the 3D image.

**Figure 9 micromachines-15-00946-f009:**
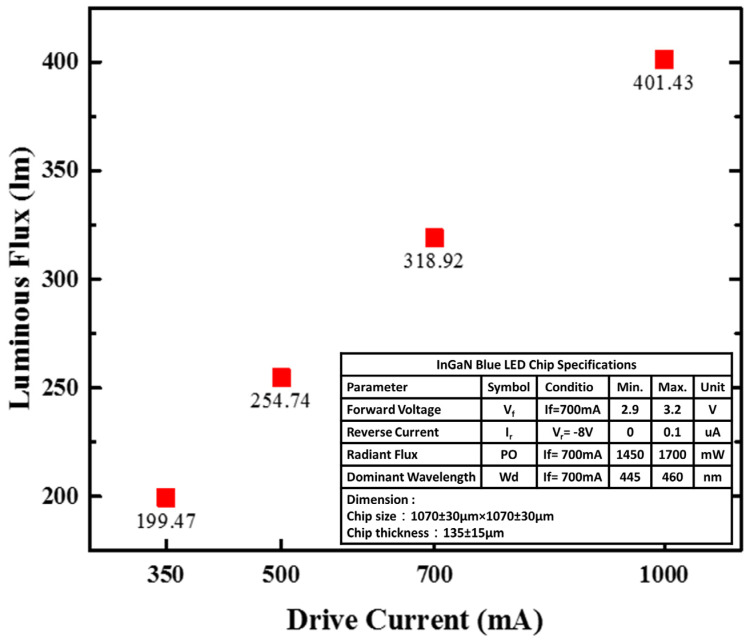
Luminous flux power of the CSP-LED.

**Figure 10 micromachines-15-00946-f010:**
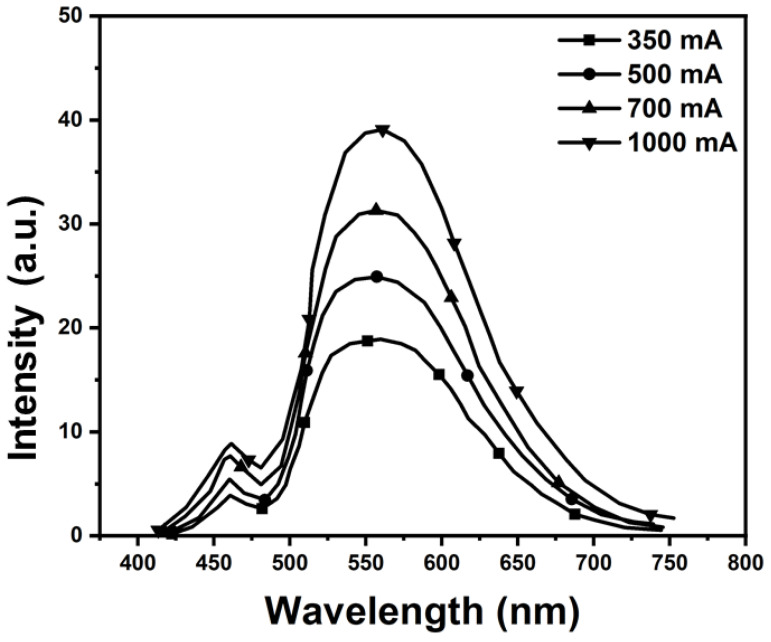
Spectral luminous efficiency of the CSP-LED.

**Figure 11 micromachines-15-00946-f011:**
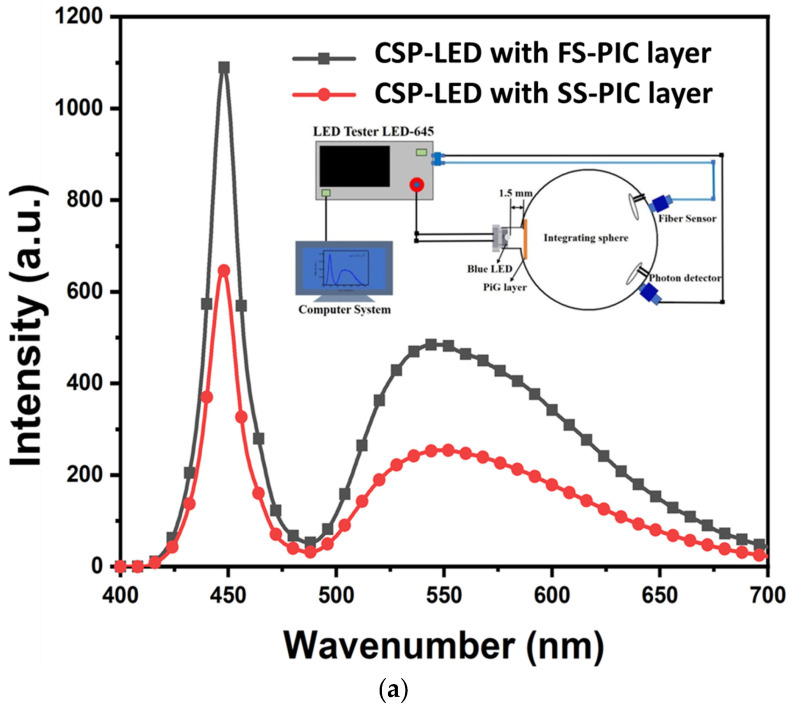
(**a**) The spectral luminous efficiency, (**b**) CIE chromaticity coordinates of the CSP-LED without/with FS-PiG layer. As illustrated in (**a**), the measurement setup employs the LED Tester LED-645 from the ISM-360 series by ISUZU OPTICS to supply the driving current to the blue LED. A 450 nm GaN-based blue LED serves as the excitation light source. The experimental configuration includes an integrating sphere testing system with optical fibers and a spectrometer to record the spectral data. This system is used to evaluate the photoluminescence spectrum, CIE 1931 (x, y) chromaticity coordinates, correlated color temperature (CCT), color rendering index (CRI), and photoluminescence efficiency of the Phosphor-in-Glass (PiG) samples. and (**c**) the structure of CSP-LED with SS-PIC layer and FS-PIC layer, respectively.

**Figure 12 micromachines-15-00946-f012:**
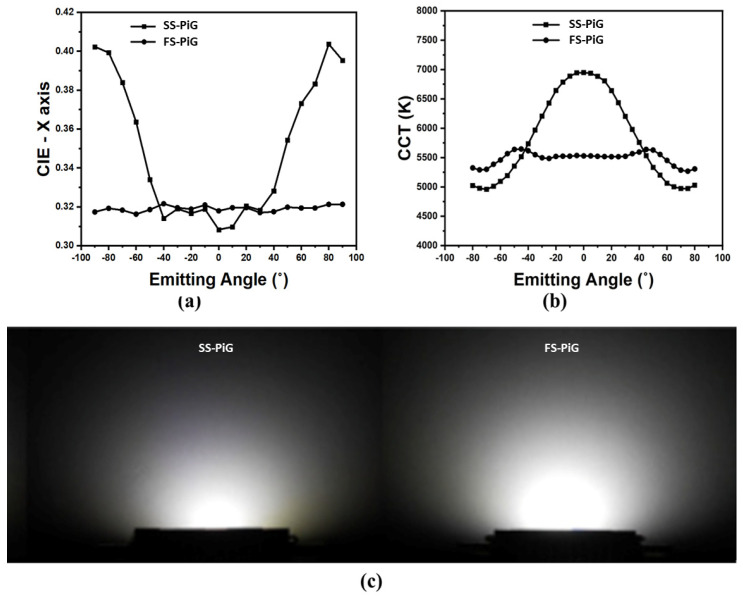
The optical measurement in the (**a**) X-axis profile distribution in the CIE, (**b**) CCT, and (**c**) emitting area of the FS-PiG and the SS-PiG.

**Figure 13 micromachines-15-00946-f013:**
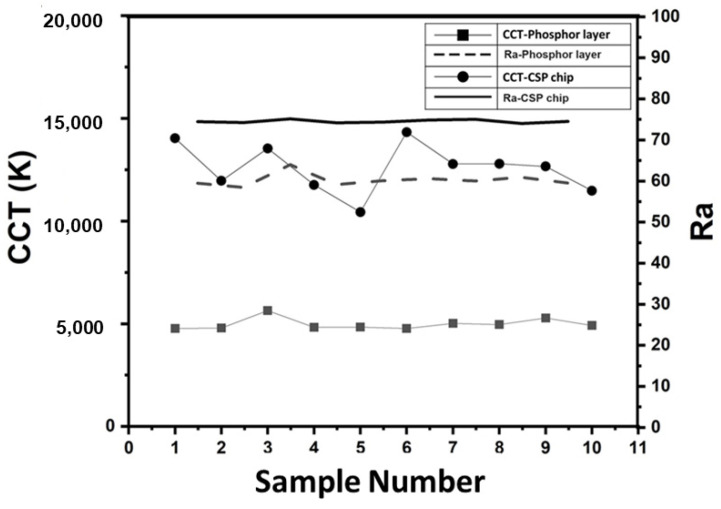
The CCT and Ra of spectral power distribution in every 10 samples, which samples were a flat fluorescent sheet on a blue chip and a fluorescent sheet with saddle structure on a blue light chip.

**Figure 14 micromachines-15-00946-f014:**
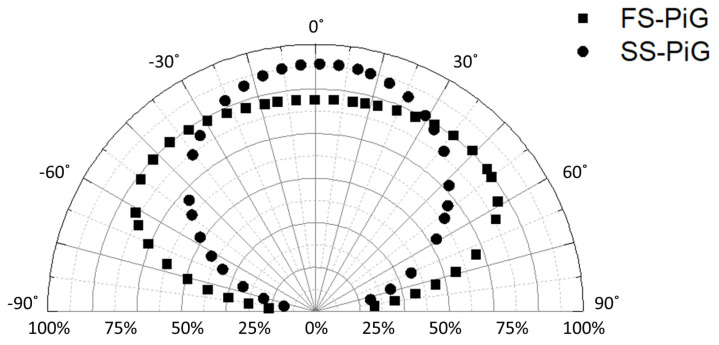
The light distribution curves of FS−PiG and SS−PiG, encapsulated respectively with blue LEDs, were measured and analyzed using a goniophotometer.

**Figure 15 micromachines-15-00946-f015:**
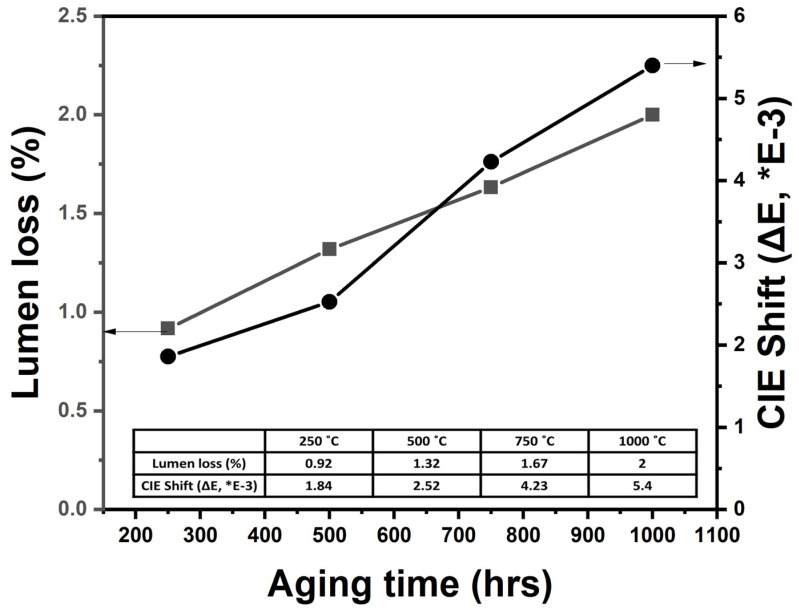
The test result of lumen loss and CIE shift when FS-PiG is 350 °C and passes to 10,008 h.

**Table 1 micromachines-15-00946-t001:** Picosecond laser parameters.

Parameter	
Wavelength	1064 nm
Laser power	44 W
Pulse repetition rate	350 kHz
Focal spot diameter	13 μm to 15 μm
Pulse width	15 ps
Cutting speed	1300 mm/s

## Data Availability

The original contributions presented in the study are included in the article, further inquiries can be directed to the corresponding authors.

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
