# Peer review of "Five-Surface Phosphor-in-Glass for Enhanced Illumination and Superior Color Uniformity in Large-View Scale LEDs"

_micromachines, 2024, doi:10.3390/mi15080946_

Round 1

Reviewer 1 Report

Comments and Suggestions for Authors

This paper demonstrates a novel five-surface phosphor-in-glass (FS-PiG) for high illumination and excellent color uniformity in larger-scale LEDs for sensor light source application. However, some parts are not clear enough and this paper did not show the large-scale LEDs. We think it should be major revised before publishing.

Comments:

1.)  Normally, it is not suitable to use words like “novel” or “new” in the title, because the novelty of the achievement in your paper could just be kept during a period.

2.)  Please clarify that Figure 1d is the absorption or excitation spectrum. How did you measure it?

3.)  In the Fabrication part, the ratio of the B2O3, Sb2O3, SiO2, and Ta2O5 should be illustrated.

4.)  Scale bars should be provided in the images of Figures 4, 5, 6, 7, and 8.

5.)  As shown in Figure 11, the LED without FS-PIG still also has the yellow emission. How did it come? It should be cleared.

6.)  What is the structure of the SS-PIG you used in Figure 12?

7.) Most important, your title said the “color uniformity in large scale LEDs”, while the LEDs used in this paper are in common size. We can’t find the large-scale one.

Comments on the Quality of English Language

Minor editing of the English language required.

Author Response

Dear Committee Members:

                                           Please find attached our team's response to your comments.

                                           Sincerely,

                                                                                                                                      Chien-Wei Huang

Reviewer 2 Report

Comments and Suggestions for Authors

In this manuscript, a sophisticated laser engraving was employed to develop a novel five-surface phosphor-in-glass (FS-PiG) for high illumination and excellent color uniformity in larger-scale LEDs for sensor light source application. The resulting FS-PiG LEDs exhibited the illumination of 401lm, average color temperature (CCT) of 5488K±110K, and color coordinates (CIE) of (0.3179±0.003, 0.3352±0.003), which are better than the single-surface-PiG LEDs. The results are interesting. I recommend its publication after addressing the following two concerns.

1.  Will the laser engraving significantly increase the cost of PiG? Is it possible for mass-production ?

2. If we use 3D printing to fabricate five-surface PiG (see Nature Communications 2020, 11, 2805),  the procedures will be cost-effective and controllable? this point should be commented or compared to show the advantages of the laser engraving.

Author Response

(The authors gave the same response as above.)

Round 2

Reviewer 1 Report

Comments and Suggestions for Authors

Most comments are well-revised. However, the previous comment 7 revision about "large scale" should be further improved. Giving more evidence to support the opinion will make it more convincing.